# Prognostic Gene Expression-Based Signature in Clear-Cell Renal Cell Carcinoma

**DOI:** 10.3390/cancers14153754

**Published:** 2022-08-01

**Authors:** Fiorella L. Roldán, Laura Izquierdo, Mercedes Ingelmo-Torres, Juan José Lozano, Raquel Carrasco, Alexandra Cuñado, Oscar Reig, Lourdes Mengual, Antonio Alcaraz

**Affiliations:** 1Laboratori i Servei d’Urologia, Hospital Clínic de Barcelona, 08036 Barcelona, Spain; flroldan@clinic.cat (F.L.R.); lizquier@clinic.cat (L.I.); ingelmo@clinic.cat (M.I.-T.); racarrasco@clinic.cat (R.C.); scunado@clinic.cat (A.C.); aalcaraz@clinic.cat (A.A.); 2Genètica i Tumors Urològics, Institut d’Investigacions Biomèdiques August Pi i Sunyer (IDIBAPS), 08036 Barcelona, Spain; 3Plataforma de Bioinformàtica, Centro de Investigación Biomédica en Red Enfermedades Hepáticas y Digestivas (CIBERehd), Hospital Clínic, 08036 Barcelona, Spain; juanjo.lozano@ciberehd.org; 4Servei d’Oncologia Mèdica, Hospital Clínic de Barcelona, 08036 Barcelona, Spain; oreig@clinic.cat; 5Departament de Biomedicina, Facultat de Medicina i Ciències de la Salut, Universitat de Barcelona (UB), 08036 Barcelona, Spain

**Keywords:** gene expression, clear-cell renal cell carcinoma, disease progression, prognostic factors, biomarkers, RNA sequencing

## Abstract

**Simple Summary:**

In this study, we identified molecular markers for disease progression from ccRCC tissue samples. Using the selected biomarkers and clinical data from the TCGA cohort, we developed a gene expression-based signature which enhances the prognostic prediction of clinicopathological variables and could help to provide personalized disease management.

**Abstract:**

The inaccuracy of the current prognostic algorithms and the potential changes in the therapeutic management of localized ccRCC demands the development of an improved prognostic model for these patients. To this end, we analyzed whole-transcriptome profiling of 26 tissue samples from progressive and non-progressive ccRCCs using Illumina Hi-seq 4000. Differentially expressed genes (DEG) were intersected with the RNA-sequencing data from the TCGA. The overlapping genes were used for further analysis. A total of 132 genes were found to be prognosis-related genes. LASSO regression enabled the development of the best prognostic six-gene panel. Cox regression analyses were performed to identify independent clinical prognostic parameters to construct a combined nomogram which includes the expression of *CERCAM*, *MIA2*, *HS6ST2*, *ONECUT2, SOX12, TMEM132A,* pT stage, tumor size and ISUP grade. A risk score generated using this model effectively stratified patients at higher risk of disease progression (HR 10.79; *p* < 0.001) and cancer-specific death (HR 19.27; *p* < 0.001). It correlated with the clinicopathological variables, enabling us to discriminate a subset of patients at higher risk of progression within the Stage, Size, Grade and Necrosis score (SSIGN) risk groups, pT and ISUP grade. In summary, a gene expression-based prognostic signature was successfully developed providing a more precise assessment of the individual risk of progression.

## 1. Introduction

Renal cell carcinoma (RCC) ranks third among the urological cancers with the highest incidence. Over 431,000 new cases and more than 170,000 RCC-related deaths were reported worldwide last year [1]. Clear-cell RCC (ccRCC) is the most common histological subtype and has the worst prognosis among all RCCs [2]. Currently, most of the newly diagnosed ccRCC cases are organ-confined tumors; however, after curative treatment, up to 30–40% will develop tumor metastases. Unfortunately, metastatic patients have a very poor five-year survival rate, varying between 0–20% [3,4]. 

According to the European Urological Guidelines, the standard management for all localized ccRCCs receiving surgery is limited to radiographic surveillance. No adjuvant treatment is approved for patients with a higher risk of progression [2]. Moreover, all surveillance recommendations are only based on clinical parameters, even though these have been proven insufficient to accurately predict disease progression, either to select ccRCC patients for adjuvant treatments or guide disease management [5,6].

Gene expression profiling has been used extensively in cancer research and has led to the discovery of new molecular prognostic markers and potential therapeutic targets. Several genetic models have been proposed in ccRCC [7,8,9,10]; however, none of those classifiers have been widely accepted nor implemented in routine clinical practice. Biomarker research for ccRCC still faces multiple challenges, mainly due to tumor heterogeneity and lack of validation studies. In addition, the use of high-throughput assays and the identification of a significant number of markers in a relatively small number of patients increase the complexity of data analysis [3]. The currently validated gene signatures comprise a large number of biomarkers, hindering their applicability and reproducibility [8,9]. Therefore, in this study we sought to develop a novel and high-performing gene expression-based signature using data generated from our cohort and The Cancer Genome Atlas (TCGA) cohort, to provide a more accurate assessment of the individual risk of progression for patients with localized ccRCC. 

## 2. Materials and Methods

### 2.1. Patients, Datasets Sources and Study Design

This study was split into a three-stage approach: an initial molecular profiling, a selection and verification of prognosis-related genes and a signature development phase (Figure 1). The initial molecular profiling phase included a total of 26 localized ccRCCs who underwent partial or radical nephrectomy between 2001 and 2010 in our center (Hospital Clinic of Barcelona, Barcelona, Spain). These 26 cases consisted of 13 progressive and 13 non-progressive patients and met the following criteria: no neoadjuvant or adjuvant treatment, no prior or concomitant malignancies or a history of inherited von Hippel-Lindau disease, all patients had thoracoabdominal CT scan staging within two month before surgery to ensure organ-confined disease. Tumors were considered progressive when local relapse or distant metastasis developed during follow-up. All patients were followed up postoperatively according to the European Urology guidelines. Any progressive patient within two months of surgery was excluded from the study. Non-progressive patients had a minimum follow-up of 10 years to ensure their status as appropriate controls. Tissue samples were obtained under institutional review board-approved protocols (HBC/2016/0333).

The selection of prognosis-related genes and signature development phases was carried out using The Cancer Genome Atlas (TCGA) dataset. Level 3 RNAseq expression data and the corresponding clinical data from TCGA ccRCC samples were obtained from the portal (https://firebrowse.org (accessed on 8 October 2021) [11] (Appendix A). Survival data were obtained from portal (https://www.sciencedirect.com/science/article/pii/S0092867418302290?via%3Dihub#app2 (accessed on 8 October 2021) [12]. After selecting samples matching our selection criteria and excluding patients without survival status or missing clinical data, a total of 356 ccRCC samples from TCGA, 68 progressive and 288 non-progressive, were selected and the gene expression of 20,532 genes was downloaded. 

### 2.2. Tissue Specimens and RNA Isolation

Formalin-fixed paraffin-embedded (FFPE) tissue blocks were reviewed. The tumor area was macro-dissected from slides (total thickness 80 µm) and RNA was isolated from FFPE specimens using the kit RecoverAll™ Total Nucleic Acid Isolation for FFPE (Ambion, Inc. Austin, TX, USA), following manufacturers’ instructions. RNA was quantified by spectrophotometric analysis at 260 nm (NanoDrop Technologies, Wilmington, DE, USA) and RNA integrity was assessed using Agilent 2100 Bioanalyzer System. 

### 2.3. Molecular Profiling by RNA Sequencing

Library preparation and sequencing method: Following rRNA removal (Ribo-Zero^®^ rRNA Removal Kit, Illumina), RNA from 26 selected ccRCC samples was processed for library preparation using the TruSeq^®^ RNA Access Library Preparation Kit (Illumina, San Diego, CA, USA) that allows generating libraries starting from degraded RNA. Briefly, cDNA strands were synthetized from input RNA in order to be adaptor-tagged, labeled and amplified. cDNA was then pooled and enriched by a double step of probes hybridization. The enriched targets were captured by streptavidin labeled beads, cleaned up and amplified to obtain the final multiplexed libraries. The libraries were then sequenced on an Illumina HiSeq^®^ 4000 platform (Illumina^®^).

Read alignment and differential gene expression analysis: Paired-end RNA-Seq FASTQ files were trimmed from a 3′ end to a fixed length based on the Phred quality score (trimmed if score fell below 20, with a minimum read length of 25) [13]. Trimmed RNA-seq reads were aligned to the GRCh38 reference genome with STAR [14] and gene counts were determined using quantMode GeneCounts. Trimmed reads were then aligned using STAR. We used limma-voom transformation and cyclic-loess to normalize the non-biological variability. An assessment of differential expression between groups was evaluated using moderated t-statistics [15].

Significant DEGs between progressive and non-progressive patients were identified based on an adjusted *p*-value of <0.05 and a fold change (FC) ≥ ±2. The heatmap and statistical analyses were performed using the R statistical package (v3.3.2). Gene set enrichment analysis (GSEA) was performed using GSEA2-2.2.0 software for testing specific gene sets based on Gene Ontology (GO) Biological Processes [16]. The “EnrichmentMap” plug-in of Cytoscape was used to create an enrichment map of the GSEA results, depicting the overlap among pathways, with similar biological processes grouped together as subnetworks [17,18,19]. A conservative overlap coefficient (0.5) was used to build the enrichment map. The “AutoAnnotate” plug-in identified clusters in an automated manner, visually annotating them with a summary label [20]. RNAseq files and clinical information were deposited in the Gene Expression Omnibus (GEO) with accession number GSE175648.

### 2.4. Selection and Verification of Prognosis-Related Genes

The DEGs identified in the previous phase were intersected with the TCGA gene expression dataset and overlapping genes were used for further analysis. The raw counts of RNA-sequencing data from the TCGA cohort were normalized using log2-based transformation. This normalized expression was used to build a multigene signature panel. Firstly, we performed univariate Cox regression analysis (considering 637 genes and 3 clinical variables) to identify the potential prognosis-related variables. Then, LASSO (Least Absolute Shrinkage and Selection Operator) regression was applied to construct the gene-based signature for predicting tumor progression in ccRCC using the cvr.glmnet function from ipflasso R package using ten-fold cross-validation and repeated it ten successive runs to increase reliability and robustness [21]. In the machine learning procedure, we fixed the three clinical variables. For all statistical analysis, a *p*-value < 0.05 was considered significant. All statistical analyses were performed using SPSS 19.0 (Statistical Product and Service Solutions; IBM Corporation, Armonk, NY, USA) and R version 3.4.2 (R Foundation for Statistical Computing, Vienna, Austria).

### 2.5. Development of a Gene Expression-Based Signature 

Based on the expression of each gene discovered and the three clinical variables, each patient’s risk score (RS) was calculated according to the risk score model. The risk score model was then used to evaluate the ccRCC prognosis according to the general form RS = exp *Σβixis*, where *i* = 1, *k* index variables, *βi* represents the coefficient for each variable estimated from the Cox regression model, and *xis* the corresponding value for each variable in a given patient. RS was subjected to a Receiver Operating Characteristics (ROC) curve analysis to choose the most appropriate threshold for predicting tumor progression. Thereafter, Kaplan–Meier curves were generated using the selected cut-off point and compared according to the log-rank test. 

The endpoints were disease progression, defined as any local relapse or distant metastasis demonstrated by radiological imaging, and cancer-specific survival. We investigated the role of the gene panel alone, clinicopathological variables alone, and a combined model including gene expression and clinicopathological variables as potential predictors of disease progression and cancer-specific survival. 

### 2.6. Pathway Enrichment Analysis

Ingenuity Pathway Analysis (IPA) software was used to identify interactions and networks between the prognostic markers included in our gene signature, possible altered canonical pathways, regulators, diseases and functions based on direct/indirect and experimental targets.

## 3. Results

### 3.1. Clinical Features of the Cohort

The clinicopathological characteristics of patients divided by study phase are summarized in Table 1. The median (range) follow-up of the cohort was 45.6 (2.1–135.8) months. During the follow-up period, a total of 68 patients (19.1%) developed tumor progression and a total of 36 patients died of ccRCC. The median time to tumor progression and cancer-related death was 19.9 (2.1–125.5) and 48.5 (2.5–151.2) months, respectively.

### 3.2. Molecular Profiling of ccRCC Samples

Overall, we identified 1380 transcripts that were differentially expressed (*p* < 0.05) between progressive and non-progressive ccRCC samples. Of these, 639 were protein-coding genes; 217 were downregulated and 422 upregulated in progressive compared with non-progressive cases. A heat map based on the most DEGs between the two groups of ccRCC patients is shown in Figure 2A. Gene set enrichment analysis (GSEA) identified several enriched biological processes, such as the dependent toll-like receptor signaling pathway, metabolic process and immune response regulating cell surface receptor signaling pathway (Figure 2B). The full list of GO biological processes is available in Appendix A. To aid interpretation of these enriched pathways, we used enrichment maps to create a network-based representation of our results. The most prominent cluster of significantly enriched pathways recapitulated changes in the Catabolism Biological Process and Toll Signaling Pathway (Figure 2C).

### 3.3. Identification of Prognosis-Related Genes in an External Data Set

To validate the 639 genes identified as DEGs in the previous study phase, these genes were intersected with the 20,532 genes from the TCGA cohort (Figure 1). As a result, we obtained 637 overlapping genes. Of those, univariate Cox regression analysis identified 132 prognosis-related DEGs and three clinicopathological variables (pT stage, tumor size and ISUP grade). 

LASSO regression analysis was used to select the best combination of genes significantly associated with disease progression and to build a six-gene signature. The expressions of five of these genes: *CERCAM*, *HS6ST2*, *ONECUT2*, *SOX12* and *TMEM132A* were upregulated, while *MIA2* expression was downregulated in progressive compared with non-progressive cases. According to Ingenuity Pathway Analysis (IPA), these six validated genes were enriched in cancer, organismal injury, abnormalities, cell-to-cell signaling and interactions, cell-mediated immune response, cellular development, cellular growth and proliferation, carbohydrate metabolism, and angiogenesis, among others. Significant IPA canonical pathways are depicted in Appendix A. The network generated shows that there were no direct interactions between the six prognostic genes (Appendix A).

Gene expression values for each selected gene were used for Cox regression analysis. High expression of *CERCAM*, *HS6ST2*, *ONECUT2*, *SOX12* and *TMEM132A* and low expression of *MIA2* related to poor outcomes for progression-free survival (Appendix A) and cancer-specific survival (Table 2). Moreover, the clinicopathological variables pT stage, tumor size and ISUP were also found as to be prognostic factors for both survival endpoints.

### 3.4. Development of a Prognostic Signature

The risk score (RS) for disease progression was calculated for each patient according to a mathematical algorithm containing the six-gene expression values; pT stage, tumor size and ISUP grade (Appendix A). An ROC analysis of this combined gene expression–clinicopathological model was performed and allowed the selection of a threshold of 0.789 (sensitivity 90% and specificity 60%) and 0.799 (sensitivity 94% and specificity 55%) to categorize patients into high- and low-risk groups for tumor progression and cancer-related death, respectively. The Kaplan–Meier curve of the combined generated gene expression-based model was able to discriminate two groups with significantly different probabilities of tumor progression (hazard ratio (HR) 10.79; 95%, *p* < 0.001) and cancer-specific survival (HR 19.27; 95%, *p* < 0.001) (Figure 3). Notably, the performance of the combined gene expression-based model (Area under the Curve [AUC] 0.824) was higher than that of clinicopathological variables alone (AUC 0.766) or gene expression data alone (AUC 0.753) (Appendix A). 

### 3.5. Correlation Analysis of the RS with Clinical Characteristics for Disease Progression

Given the clinical significance of the RS in ccRCC, we sought to investigate the potential correlation between RS and clinical features. The Mann–Whitney test revealed that higher RSs correlated with a higher risk group within the SSIGN model, higher pT stage and higher ISUP grade (Figure 4). Furthermore, the Kaplan–Meier curve indicated that our established RS was capable of identifying ccRCC patients at the highest risk of progression within the groups stratified by SSIGN, pT stage and ISUP grade (all *p* < 0.05; Appendix A).

## 4. Discussion

Currently, clinicopathological variables are the most valuable tool for predicting disease outcomes in ccRCC. However, due to the highly variable behavior of ccRCC, the prediction of tumor progression is still an important clinical challenge. For decades, surgery has remained the only treatment approved for localized ccRCC [2]. At present, disease management of these patients at high risk of progression is changing and new adjuvant treatments are being considered, opening a door to more personalized medicine in localized renal carcinoma [23,24]. 

Molecular profiling helps our understanding of the molecular mechanism underlying ccRCC and affords great potential to identify new biomarkers of clinical utility [25,26]. However, intratumor heterogeneity and the methodology for sample processing, readout and expression normalization have been a strong challenge in the development of a robust gene signature. Next-generation sequencing is the most advanced technique for gene expression profiling [27]; here, we used this technology to analyze the entire transcriptome profiling of tissue samples from progressive and non-progressive ccRCC patients. We then established a gene signature for predicting disease progression based on the gene expression and clinical data obtained from the TCGA cohort. We improved the gene selection and accuracy of the model by using LASSO regression analysis; this allowed us to include all DEGs found in the training set and avoid the preselection and validation of only a subset of these DEGs [28]. 

This study demonstrated that our gene expression-based signature was able to identify localized ccRCC patients with high and low risk of disease progression in the whole cohort and within the SSIGN risk groups. It properly correlated with clinical parameters and was proven to enhance the predictive value of the current clinicopathological variables. Furthermore, it was also predictive of cancer-specific survival. Therefore, our signature may constitute an important step forward in treatment decisions for ccRCC patients.

Remarkably, the developed gene-based panel demonstrated a greater value for prognostic prediction (HR 10.79, *p* < 0.001; AUC = 0.824) compared with similar, previously described models. Dai et al. proposed a four-gene signature with an HR of 3.1, *p* < 0.001, whereas Zhao et al. described a 15-gene model with an AUC of 0.737 [29]. Unfortunately, the different designs and methodologies of several other studies thwart any performance comparisons with their proposed genetic models. Thus, Brook et al. [9] assessed the performance of ClearCode34, which classifies ccRCC into ccA and ccB subtypes, ccB presented tumor relapse more frequently (HR 2.1; *p* = 0.001), whereas Rini et al. [8] generated a 16-gene signature associated with tumor recurrence with an HR per 25-unit increase in score of 3.37 (*p* < 0.001). Likewise, other authors have built molecular signatures aiming to predict overall survival (OS), thus making the classifiers’ performance non-comparable [30,31,32]. 

Biologically, the genes from our panel are unrelated to each other and many of them have been shown to have either prognostic or biologic relevance in tumor metastasis development. According to previous reports, some of our selected genes were consistent with previously discovered biomarkers; therefore, we have further validated their value for ccRCC progression. Briefly, *CERCAM* (cerebral endothelial cell adhesion molecule) is an adhesion molecule found to be an unfavorable prognostic marker in several tumors [33,34,35]. Its overexpression promotes cell viability, proliferation and invasion, it is involved in the PI3K/AKT pathway [36] and is part of an immune prognostic signature for colon and rectal cancer [35]. *HS6ST2* (heparan sulfate D-glucosaminyl 6-*O*-sulfotransferase-2) is a glycolysis-related gene and has also been associated with poor disease outcomes in numerous malignancies [10,37]. Interestingly, our group previously validated this gene as an independent prognosis biomarker in intermediate/high-risk ccRCC and found it to be associated with angiogenesis, epithelia–mesenchymal transition (EMT), and indirectly related to the PD-1, PDL-1 cancer immunotherapy pathway [37,38,39]. *MIA2* (melanoma inhibitory activity 2) has been found in several malignancies and can act as a tumor suppressor [40] or as a proto-oncogene depending on the receptor-related signaling differences [41]. We found *MIA2* to be downregulated in progressive ccRCC. This is congruent with the human protein atlas findings, where high expression was a favorable prognostic factor in renal cancer [33,42]. *ONECUT2* (One cut domain family member 2) is a transcription factor able to activate oncogenic pathways and lineage-specific genes; hence, it is involved in EMT, angiogenesis, neural differentiation, proliferation, extracellular matrix organization, cell locomotion and migration, among others. Overexpression of *ONECUT2* has been described in several tumors and is related to poor prognosis [43,44,45,46]. *SOX12* (Sex-determining region Y-box12) is a transcription factor, its upregulation promotes tumor progression and it is involved in EMT, apoptosis and cell proliferation [47,48,49]. It functions as an oncogene-regulating Wnt/B-catenin signaling to promote the growth of multiple myeloma cells [50]. As for *TMEM132A*, few studies were found in the literature, so further investigations are required to establish its role in tumor development. 

Our study has multiple strengths. The first advantage of our gene expression-based signature is that it contains a low number of genes, making its clinical application easier. Our model did match genes from previous models and some of them exceeded the mere field of ccRCC, highlighting the prognostic power of the selected genes. The fact that they are involved in different carcinogenic mechanisms confers an advantage to our signature compared with others that only target one single pathway [10,51]. The high-throughput technology used to analyze the samples and the statistical methodology makes our gene model a reliable tool for predicting disease progression in ccRCC and adds important prognostic information to the clinicopathological parameters. However, we acknowledge that this study has several limitations. First, the retrospective design and the relatively small sample might have influenced our findings. Second, the definition of CSS in the TCGA cohort should be taken with caution since it was estimated [12]. Finally, despite the good performance of our six-gene model, further validations in larger cohorts are required.

## 5. Conclusions

A gene expression-based prognostic signature to predict disease progression in ccRCC was successfully developed; it could discriminate two groups with different probabilities of tumor recurrence. In addition, our model was also useful in predicting cancer-specific survival. The combination of genetic and clinical information enhanced the current risk stratification of the localized ccRCC patients. Refining prognostic algorithms could help to improve the disease management and follow-up of ccRCC patients.

## Figures and Tables

**Figure 1 cancers-14-03754-f001:**
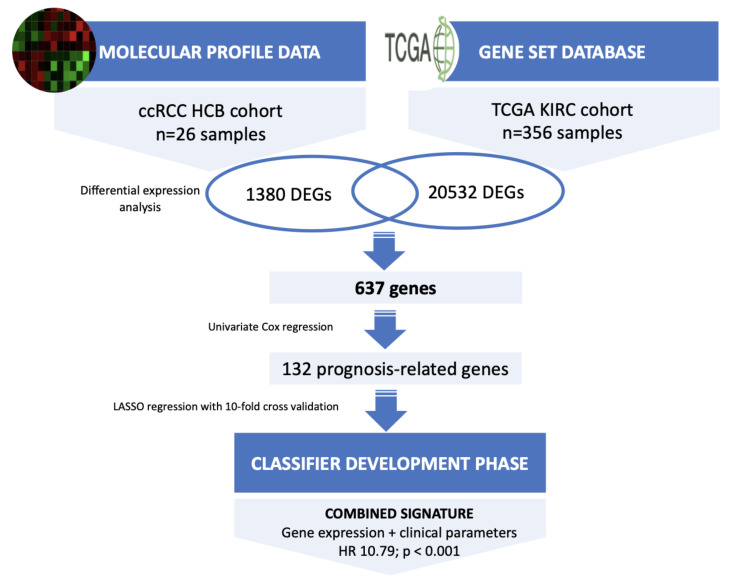
Flowchart of the whole study. Abbreviations: ccRCC, clear-cell renal cell carcinoma; DEGs, differentially expressed genes.

**Figure 2 cancers-14-03754-f002:**
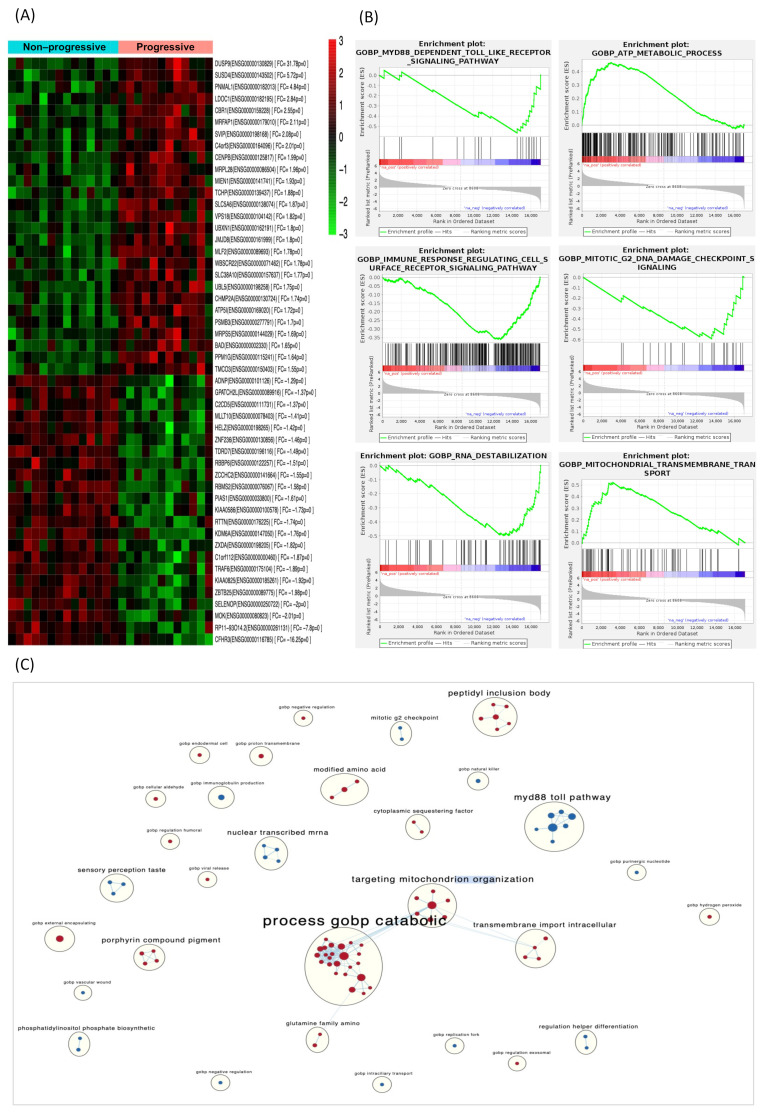
DEGs in the discovery phase and gene-set enrichment analysis. (**A**) Heat map displaying the 50 most DEGs between progressive and non-progressive localized ccRCC patients. Red pixels correspond to upregulated genes, whereas green pixels indicate downregulated genes. (**B**) GSEA shows positive correlation of DEGs in biological processes involved in tumor progression. (**C**) Enrichment map where nodes represent gene sets (pathways) and edges (blue lines) denote overlapping genes between 2 pathways. Node size denotes gene set size. Predicted pathways are grouped as circles, where shades in red correspond to up-regulated gene-sets and shades in light blue correspond to down-regulated gene-sets. Highly redundant gene sets are grouped together as clusters. Abbreviations: DEGs, differentially expressed genes. GSEA, gene set enrichment analysis.

**Figure 3 cancers-14-03754-f003:**
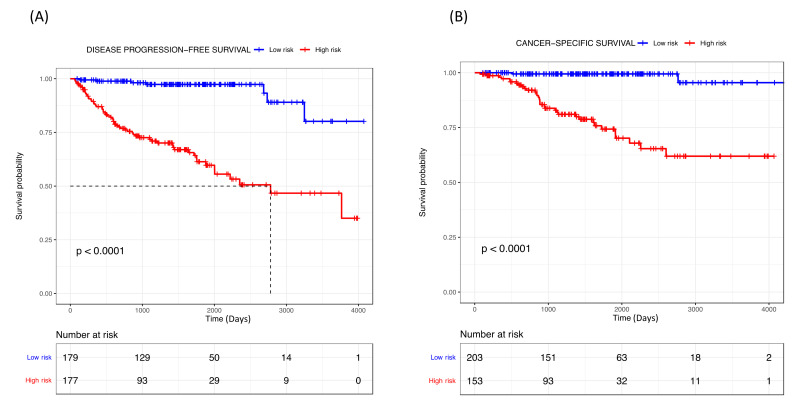
Kaplan–Meier curves of the combined gene expression-based model for (**A**) disease progression-free survival and (**B**) cancer-specific survival for TCGA cohort.

**Figure 4 cancers-14-03754-f004:**
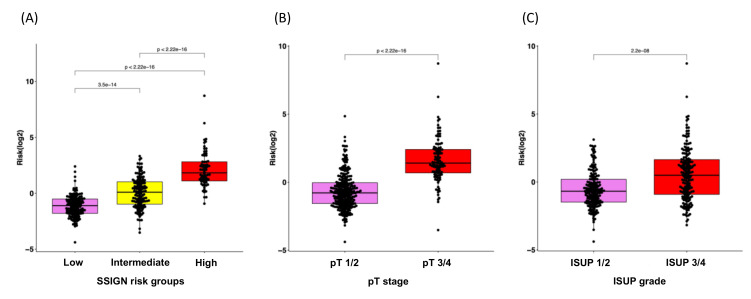
Box plots for the correlation analysis of RS with clinical characteristics for disease progression. (**A**) SSIGN risk groups, (**B**) pT stage and (**C**) ISUP grade.

**Table 1 cancers-14-03754-t001:** Demographic and pathological characteristics of enrolled patients.

KERRYPNX	Discovery Phase Hospital Clinic Barcelona (*n* = 26)	Validation Phase TCGA Cohort (*n* = 356)
Gender		
Male	18 (69.2)	231 (64.9)
Female	8 (30.8)	125 (35.1)
Age at diagnosis (year)	59 (34–81)	60 (29–90)
Pathological tumor size (cm)	5.5 (1.9–17.5)	5.1 (1.0–25)
ISUP		
ISUP 1	3 (11.5)	4 (1.1)
ISUP 2	12 (46.2)	173 (48.6)
ISUP 3	6 (23.1)	145 (40.7)
ISUP 4	5 (19.2)	34 (9.6)
Tumor stage		
pT1	15 (57.7)	211 (59.3)
pT2	5 (19.2)	41 (11.5)
pT3	5 (19.2)	102 (28.7)
pT4	1 (3.8)	2 (0.6)
N stage		
N0/x	24 (92.3)	346 (97.2)
N1	2 (7.7)	10 (2.8)
Necrosis	10 (38.5)	144 (40.4)
SSIGN score *		
Low risk	12 (46.2)	143 (40.2)
Intermediate risk	8 (30.7)	141 (39.6)
High risk	6 (23.1)	72 (20.2)

* Stage, Size, Grade and Necrosis (SSIGN) score [22].

**Table 2 cancers-14-03754-t002:** Univariate Cox regression analysis of statistically significant genetic and clinical variables in the validation set (TCGA cohort).

	Progression-Free Survival			Cancer-Specific Survival		
	*p*	95% CI	HR	*p*	95% CI	HR
*CERCAM*	<0.001	1.387–3.807	2.298	<0.001	1.036–1.075	1.055
*HS6ST2*	<0.001	1.164–3.106	1.902	0.034	1.043–2.866	1.729
*MIA2*	<0.001	0.222–0.632	0.375	<0.001	0.825–0.935	0.878
*ONECUT2*	0.015	1.111–2.952	1.811	<0.001	2.443–5.942	3.810
*SOX12*	0.001	1.354–3.748	2.252	<0.001	1.177–1.488	1.323
*TMEM132A*	<0.001	1.526–4.288	2.558	<0.001	1.070–1.156	1.112
pT Stage	<0.001	1.775–3.024	2.317	<0.001	2.547–9.940	5.032
Tumor size	<0.001	1.154–1.273	1.212	<0.001	1.125–1.271	1.195
ISUP	<0.001	1.568–3.158	2.225	0.001	1.628–7.845	3.574

## Data Availability

The data presented in this study are available in the article and Appendix A. RNAseq files and clinical information were deposited in the Gene Expression Omnibus (GEO) with accession number GSE175648. Further details can be obtained on request to the corresponding author.

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
