# Peer review of "Prognostic Gene Expression-Based Signature in Clear-Cell Renal Cell Carcinoma"

_cancers, 2022, doi:10.3390/cancers14153754_

Round 1
Reviewer 1 Report
Roldan et al identified molecular markers from ccRCC tissue samples. Using the selected biomarkers and clinical data from the TCGA cohort, they developed a gene expression-based signature which enhances prediction of clinico-pathological variables and could help to provide personalized disease management.
The gene signature they identified seemed to perform well in prognosis in the larger TCGA cohort. It will be nice to generate the Kaplan Meier curves based on SSIGN, pT stage and ISUP risk groups as they did in Fig. 3, so the readers can get a sense of the relative strength of prediction by their new model. This can be presented in a supplement data file.
Their method tracks well with SSIGN, pT stage and ISUP risk groups. It is interesting that their model can still find difference within each risk group, which suggests that a combined approach is likely superior to a single method for prognosis. Their own data (Fig. S2) supports this, and this adds to the value of their study.
What is the value on the x axis in their Kaplan Meier curves? It is likely days. Please add this.
Author Response
Answers to comments of Reviewer #1
We thank Reviewer 4 for all their comments and suggestions about our manuscript.
Roldan et al identified molecular markers from ccRCC tissue samples. Using the selected biomarkers and clinical data from the TCGA cohort, they developed a gene expression-based signature which enhances prediction of clinico-pathological variables and could help to provide personalized disease management.
The gene signature they identified seemed to perform well in prognosis in the larger TCGA cohort. It will be nice to generate the Kaplan Meier curves based on SSIGN, pT stage and ISUP risk groups as they did in Fig. 3, so the readers can get a sense of the relative strength of prediction by their new model. This can be presented in a supplement data file.
Their method tracks well with SSIGN, pT stage and ISUP risk groups. It is interesting that their model can still find difference within each risk group, which suggests that a combined approach is likely superior to a single method for prognosis. Their own data (Fig. S2) supports this, and this adds to the value of their study.
As the reviewer suggest, this is an interesting data. Actually, we had already added this information in supplementary Figure S4. Also, we have made a new supplementary figure (Figure S2) showing the strength of prediction for each of the genes of the signature (Klaplan Meier curves).
What is the value on the x axis in their Kaplan Meier curves? It is likely days. Please add this.
Yes, it is days. We have added this label to all Kaplan Meier curves.

Reviewer 2 Report
I only identify here some minor points I would love the authors to address:
· The authors derived their samples from paraffin-embedded biopsies that were obtained more than 10 years ago. What controls were undertaken to check if the conservation and the storage were performed correctly and there was no deterioration of sample quality?
· In chapter 3.2 the authors state that they have used a p value of 0.05. I am wondering if this threshold is sensitive enough. Would the authors still identify their proposed six markers if they lower the p value to e.g. 0.001?
· Is there any chance to use the six identified markers and screen for prospective ccRcc patients that do not have any kidney issues yet? Can this expression signature be inherited? Maybe the authors can speculate upon this more in the discussion.
· Use the proper SI terms when it comes to the abbreviation of year, months,
· Revisit the English grammar and recheck potential issues, like plural or singaular (e.g. line 103 were – was), I encourage the use of a thousand-comma (1,000) to increase readability.
· Together with the journals layouter make sure the settings are right. Quote in line 88 needs to be transformed
Author Response
Answers to comments of Reviewer #2
We thank Reviewer 3 for all their comments and suggestions about our manuscript.
I only identify here some minor points I would love the authors to address: the authors derived their samples from paraffin-embedded biopsies that were obtained more than 10 years ago. What controls were undertaken to check if the conservation and the storage were performed correctly and there was no deterioration of sample quality?
As the referee indicated, our study samples have more than 10 years. The laboratory keeps a proper storage of the formalin-fixed paraffin-embedded (FFPE) tissue samples obtained in the partial or radical nephrectomy. However, as it is known, the procedure of tissue fixation in formalin and the subsequent embedding in paraffin is very aggressive for the RNA and damage it.
First control to ascertain the RNA degradation level of our samples was to perform an Agilent 2100 Bioanalyzer analysis (data not shown). This analysis reported RNA Integrity Numbers (RIN) between 1,3 and 2,8 for our samples, which confirms the partial degradation of our RNA. Considering this level of RNA degradation, we decided to use Ribo-Zero® rRNA Removal Kit (Illumina) to remove rRNA from total RNA, thus producing RNA samples ready for library prep. The Ribo-Zero protocol offers downstream sequencing data that contains complete transcriptome of coding and noncoding RNA species (see below the link to the Ribo-Zero rRNA Removal kit Reference Guide). We have added this information regarding to RNA integrity and removal of rRNA from samples to the revised version of the manuscript (see lines 102-105).
(https://jp.support.illumina.com/content/dam/illumina-support/documents/documentation/chemistry_documentation/ribosomal-depletion/ribo-zero/ribo-zero-reference-guide-15066012-02.pdf)
In chapter 3.2 the authors state that they have used a p value of 0.05. I am wondering if this threshold is sensitive enough. Would the authors still identify their proposed six markers if they lower the p value to e.g. 0.001?
As stated by the referee, we have used a p value < 0.05 in all the statistical analysis in the paper. This threshold is widely accepted in the scientific community and we think it is appropriate to select the genes for the model. Using this p value we obtained 132 genes significative in the univariate analysis. This number of genes is appropriate to obtain different gene combinations in the multivariate analysis and ensure that the selected genes are not interdependent. On the contrary, if we had used a more restrictive threshold, e.g. p < 0.01, we would have obtained 82 significative genes in the univariate analysis and with e.g. p<0.001 we would have obtained only 30. Consequently, using a p < 0.01 five out of the six genes from our models would have been significative in the univariate analysis and with a p<0.001 only four out of the six, as shown in Table 2 of the manuscript. However, it must be clear that those genes with a p value between 0.01-0.05 are as valuable as the others. In conclusion, we think that the selected p value of 0.05 have allowed us to find an accurate signature which includes an appropriate and reasonable number of significant genes.
Is there any chance to use the six identified markers and screen for prospective ccRcc patients that do not have any kidney issues yet? Can this expression signature be inherited? Maybe the authors can speculate upon this more in the discussion.
Regarding to the possibility of screening for prospective ccRcc in healthy patients, we think this would have an uncertain result. First, this is a signature to predict progression, so we do not know how these genes are expressed in healthy samples. We neither know if the expression of these genes in healthy samples differs from their expression in malignant ones. Eventually, the signature includes some clinicopathological variables, not only 6 gene expression values, but that would be missing in healthy samples.
Regarding an inherited issue of the gene expression signature, we have to take into account that we do not inherit RNA molecules but DNA. Thus, it is not possible to inherited this gene expression signature.
Use the proper SI terms, when it comes to the abbreviation of year, months. Revisit the English grammar and recheck potential issues, like plural or singular (e.g. line 103 were – was), I encourage the use of a thousand-comma (1,000) to increase readability. Together with the journals layouter make sure the settings are right. Quote in line 88 needs to be transformed
We have sent the paper for grammar revision to a native English speaker, and we have also carefully reviewed abbreviations, commas and other typo mistakes.

Reviewer 3 Report
The author perform a feature selection exercise based on differential expression of genes in their cohort of progressive and non-progressive renal clear cell cancers samples. They use these genes to identify genes that could be associated with progression (using clinical data from TCGA). They then identify a 6 gene signature that can be used to define a risk score to stratify patient's risk of progression.
The signature is not validated in an independent dataset. The methods are poorly written and its difficult to get a comprehensive understanding of the methodology used by the authors. I cannot recommend this study for publication. My detailed concerns are listed below.
Major concerns:
1. Only half of the genes that were differentially expressed in their cohort are identified in TCGA, why is this? The methods itself has a single line on how the RNAseq data was processed, without details on what genome version was used, what transcript definitions were used. There are also no details on the library prep approach and other details that are routinely provided.
2.It's also unclear to me how the differential expression was performed? Were appropriate tools like DESeq2/limma/edgeR used or was some other test used?
3. The criteria of how progression was defined in TCGA needs to be clearly stated. Further Table 1 has a lot of clinical information about TCGA samples like size, SSIGN risk score etc. Where were these details obtained from. The paper referenced as the source of clinical details doesn't seem to have this information. I could find some of these variables upon combing through clinical metadata file on FireHose maintained by Broad Institute. The authors need to clarify state where their data came from and what variables in the original data were used.
4. What are the statistical thresholds used to define significance in the univariate cox regression analysis? in the multivariate model the authors didn't control for confounding clinical variables like age, gender, tumor stage etc. Also please clearly describe the criteria for for selection of the combined gene panel. The statement "The optimal combination for the gene panel was selected based on overall performance in multivariate logistic regression analysis" is very vague.
5. How was the risk score generated. The only explanation I find in the text is "Risk Score (RS) for disease progression was calculated for each patient according to a mathematical algorithm containing the six-gene expression values, pT stage, tumor size and ISUP grade.". This provides no real details on what exactly was done to generate the score making it impossible to access the validity of their approach.
6. The authors provide no independent validation of the score they have developed. The performance seems to be accessed by reapplying the gene panel to TCGA. considering that TCGA was used to develop this panel it is not surprising that the score works well as a differentiating measure. Unless the authors can show that their approach can be applied to independent datasets and make meaningful predictions, it's difficult to evaluate the performance as a prognostic tool.
Minor concerns:
1. Figure 1 is unclear it very difficult to read what's in these plots.
2. The pathways analysis in Fig 1 is rudimentary, little effort is made to make sense of the data and interpret their observations.
Author Response
Answers to comments of Reviewer #3
We thank Reviewer 3 for all their comments and suggestions about our manuscript.
The author perform a feature selection exercise based on differential expression of genes in their cohort of progressive and non-progressive renal clear cell cancers samples. They use these genes to identify genes that could be associated with progression (using clinical data from TCGA). They then identify a 6 gene signature that can be used to define a risk score to stratify patient's risk of progression.
The signature is not validated in an independent dataset. The methods are poorly written and its difficult to get a comprehensive understanding of the methodology used by the authors. I cannot recommend this study for publication. My detailed concerns are listed below.
-----
- Only half of the genes that were differentially expressed in their cohort are identified in TCGA, why is this? The methods itself has a single line on how the RNAseq data was processed, without details on what genome version was used, what transcript definitions were used. There are also no details on the library prep approach and other details that are routinely provided.
As indicated by the referee, in our cohort there were in total 1380 differentially expressed transcripts between progressing and non-progressing ccRCC, including coding RNAs, non-coding RNAs, pseudogenes, among others. In the TCGA database only protein-coding genes are described. Thus, if we select only the differentially expressed protein-coding genes from our cohort (according to the HUGO database), we end up with 639 DEGs, from which 637 genes were found in the TCGA. This represents 99.68% of our DEG are found in the TCGA database. We have clarified this issue in the revised version of the manuscript (see lines 185-187).
As for the brief description of the methodology, we have rewritten it and explain it in more detail. We have added details on the library prep approach and how the RNAseq data was processed. We also added details the genome version and transcript definitions used (see lines 103-111).
- How the differential expression was performed? Were appropriate tools like DESeq2/limma/edgeR used or was some other test used?
As suggested by the reviewer, we have added an explanation regarding to the tools used to identify DEG (see lines 112-119).
- The criteria of how progression was defined in TCGA needs to be clearly stated. Further Table 1 has a lot of clinical information about TCGA samples like size, SSIGN risk score etc. Where were these details obtained from? The paper referenced as the source of clinical details doesn't seem to have this information. I could find some of these variables upon combing through clinical metadata file on FireHose maintained by Broad Institute. The authors need to clarify state where their data came from and what variables in the original data were used.
The clinical data was obtained from 2 sources:
- Survival data
Jianfang Liu et al. published in April 2018 this article (Ref 12): An Integrated TCGA Pan-Cancer Clinical Data Resource to Drive High-Quality Survival Outcome Analytics. (https://doi.org/10.1016/j.cell.2018.02.052). They downloaded TCGA clinical and molecular data from the portal Genomic Data Commons (GDC) (https://gdc-portal.nci.nih.gov/legacy-archive/) and presented a curated and filtered clinical and survival outcome data as a newly integrated resource for the entire scientific community. This data can be downloaded from the following link: https://www.sciencedirect.com/science/article/pii/S0092867418302290?via%3Dihub#app2.
This developed standardized dataset is named TCGA Pan-Cancer Clinical Data Resource (TCGA-CDR) and includes four major clinical outcome endpoints: overall survival (OS), progression free interval (PFI), disease free interval and disease specific survival (DSS). They concluded that OS and PFI could be derived relatively accurately using the available data from the TCGA while DSS could only be estimated.
- Additional clinical data
Despite of the previous well-curated database regarding survival outcomes, there are some missing clinical features, such as tumor size, necrosis, nodes, where we could find in the second source, the website http://firebrowse.org/, where we found all the information from every single sample: pN, necrosis, tumor size, etc.
With all this data SSIGN risk score was calculated for each patient. The excel file, containing the data from these both sources, used in this study is shown in supplementary material Table S1. In addition, we have clarified where the data come from (see lines 86-90)
Regarding to the definition of tumor progression in the TCGA cohort, it is stated in the website where we obtained the data, named as Progression free interval (PFI):
Progression free interval (PFI) was defined as 1 (= progressive patients) for patient having new tumor event, whether it was a progression of disease, local recurrence, distant metastasis, or died with the cancer without new tumor event, including cases with a new tumor event whose type is N/A. This PFI corresponds to what we call “disease progression”.
We have added TCGA definitions for Progression free interval (PFI) and disease-specific survival (DSS) in supplementary material Table S1.
- What are the statistical thresholds used to define significance in the univariate cox regression analysis? in the multivariate model the authors didn't control for confounding clinical variables like age, gender, tumor stage etc. Also please clearly describe the criteria for selection of the combined gene panel. The statement "The optimal combination for the gene panel was selected based on overall performance in multivariate logistic regression analysis" is very vague.
The statistical threshold to define significance in the univariate Cox regression analysis and in all analyses carried on in this paper was 0.05. We have modified the manuscript to make this clear for all readers (see line 144).
Univariate analysis was performed with all 637 genes. We also added to the univariate analysis the main 3 prognostic clinical variables (pT stage, tumor size and ISUP grade). As a result, we found 132 genes and the 3 clinicopathological variables statistically significant (p <0.05). We have added this information to the manuscript (see lines 138-139 and 209-210).
To select the variables integrating the proposed gene-based signature, we performed LASSO.
LASSO works in a complex manner; however, summarizing one of its main properties is that allow us to automatically select those features that are useful, discarding the useless or redundant features. It uses this cost function:
aj is the coefficient of the j-th feature. The final term is called l1 penalty and α is a hyperparameter that tunes the intensity of this penalty term. The higher the coefficient of the aj feature, the higher the value of the cost function. So, the idea of Lasso regression is to optimize the cost function reducing the absolute values of the coefficients, discarding a feature will make its coefficient equal to 0. This works, if the features have been previously scaled, for example using standardization. In addition, α hyperparameter value must be found using a cross-validation approach.
Therefore, when LASSO program is run, automatically gives you the best combination for a prognosis model depending on the number of variables you want to include in your model. In our case, to perform LASSO we fixed the three clinical variables and allowed the machine learning procedure to select the genes integrating the model.
- How was the risk score generated? The only explanation I find in the text is "Risk Score (RS) for disease progression was calculated for each patient according to a mathematical algorithm containing the six-gene expression values, pT stage, tumor size and ISUP grade.". This provides no real details on what exactly was done to generate the score making it impossible to access the validity of their approach.
Following the recommendations of the referee we have explained the formulas of the risk score used in this paper (see lines 153-155).
We have also added the RS for each patient in our study as a supplementary material (see Table S4). It is important to understand that different values in the variables result in different RS, and so another threshold. Higher risk score, higher the risk to develop tumor progression and cancer-related death. In our paper, using the ROC to define the threshold we found a HR of 10.79; 95%, p < 0.001.
- The authors provide no independent validation of the score they have developed. The performance seems to be accessed by reapplying the gene panel to TCGA. considering that TCGA 2 was used to develop this panel it is not surprising that the score works well as a differentiating measure. Unless the authors can show that their approach can be applied to independent datasets and make meaningful predictions, it's difficult to evaluate the performance as a prognostic tool.
We agree that the signature is not validated in an independent cohort. However, we performed an internal validation of our results using ten-fold cross-validation and repeated it ten successive runs to increase reliability and robustness of the algorithm. We believe that despite of the lack of the external validation the methodology used provides reliable results. We have added this information to the manuscript (see lines 143-144).
- Minor concerns:
- Figure 1 is unclear it very difficult to read what's in these plots.
We have modified the Figure 1 to make it easier to understand.
- The pathways analysis in Fig 1 is rudimentary, little effort is made to make sense of the data and interpret their observations.
We believe that the reviewer refers to Figure 2. In Figure 2 we show correlations between DEG and the pathways of which these DEG are involved. DEG are grouped in pathways (represented as nodes); the biggest the size of the node, the most represented pathway. The pathway analysis is detailed in Table S2 and Figure S1.

Reviewer 4 Report
In a cohort of 26 tissues, the authors performed whole-transcriptome profiling and differentially expressed genes were compared with the transcriptomic data in the TCGA-KIRC cohort. Based on the analyses, they identified a gene signature and constructed a combined nomogram which includes the expression of CERCAM, MIA2, HS6ST2, ONE-26 CUT2, SOX12, TMEM132A, pT stage, tumor size and ISUP grade. A risk score generated using this 27 model effectively stratified patients at higher risk of disease progression.
The study is well conducted. My comments are as follows:
In the TCGA dataset- the available data is for overall survival how cancer-specific survival, progress-free survival data were obtained?
It is confusing: For which cohorts what endpoints were used? The endpoints stated are disease progression, defined as any local relapse or distant metastasis demonstrated by radiological imaging, and cancer-specific survival. The clinical cohort is only 26 patients. For the TCGA-KIRC dataset N and M stage data are available not local relapse or distant metastasis data or time to event for these parameters are available.
Is the identified gene signature specific for ccRCC? Kindly also analyze TCGA-KIRP data to enhance the scientific merit of the study.
For multivariate analyses related to individual markers, which clinical and pathological parameters were included? The multivariate data are shown only for the markers.
It is unclear which cohorts were used for calculating Risk Score (RS) for disease progression. Only KM-plots were used for showing the clinical significance of the RS.
It is unclear which data and which cohort was used to show disease progression-free survival and (B) cancer-specific survival in Figure 3. Multivariate analyses should be performed, as the increase in AUC is modest at best (0.82 versus 0.766). Please also include sensitivity and specificity values.
Author Response
Answers to comments of Reviewer #4
We thank Reviewer 4 for all their comments and suggestions about our manuscript.
In a cohort of 26 tissues, the authors performed whole-transcriptome profiling and differentially expressed genes were compared with the transcriptomic data in the TCGA-KIRC cohort. Based on the analyses, they identified a gene signature and constructed a combined nomogram which includes the expression of CERCAM, MIA2, HS6ST2, ONE-26 CUT2, SOX12, TMEM132A, pT stage, tumor size and ISUP grade. A risk score generated using this 27 model effectively stratified patients at higher risk of disease progression. The study is well conducted. My comments are as follows:
------
In the TCGA dataset- the available data is for overall survival how cancer-specific survival, progress-free survival data were obtained?
TCGA gene expression data was obtained from the portal firebrowse.org, as indicated in the manuscript and the clinical data that was obtained from 2 sources:
- Survival data
Jianfang Liu et al. published in April 2018 this article (Ref 12): An Integrated TCGA Pan-Cancer Clinical Data Resource to Drive High-Quality Survival Outcome Analytics. (https://doi.org/10.1016/j.cell.2018.02.052). They downloaded TCGA clinical and molecular data from the portal Genomic Data Commons (GDC) (https://gdc-portal.nci.nih.gov/legacy-archive/) and presented a curated and filtered clinical and survival outcome data as a newly integrated resource for the entire scientific community. This data can be downloaded from the following link: https://www.sciencedirect.com/science/article/pii/S0092867418302290?via%3Dihub#app2.
This developed standardized dataset is named TCGA Pan-Cancer Clinical Data Resource (TCGA-CDR) and includes four major clinical outcome endpoints: overall survival (OS), progression free interval (PFI), disease free interval and disease specific survival (DSS). They concluded that OS and PFI could be derived relatively accurately using the available data from the TCGA while DSS could only be estimated.
- Additional clinical data
Despite of the previous well-curated database regarding survival outcomes, there are some missing clinical features, such as tumor size, necrosis, nodes, where we could find in the second source, the website http://firebrowse.org/, where we found all the information from every single sample: pN, necrosis, tumor size, etc.
The excel file, containing the data from these sources, used in this study is shown in supplementary material Table S1.
It is confusing: For which cohorts what endpoints were used? The endpoints stated are disease progression, defined as any local relapse or distant metastasis, and cancer-specific survival. The clinical cohort is only 26 patients. For the TCGA-KIRC dataset N and M stage data are available not local relapse or distant metastasis data or time to event for these parameters are available.
As stated by the reviewer in the present work there are two cohorts:
- Cohort 1 which includes 26 patients from our hospital. Clinical data from these patients was obtained from our hospital clinical records and gene expression data was obtained by RNA sequencing experiments performed by our research group. In this cohort disease progression was defined as local relapse or distant metastasis demonstrated by radiological imaging and cancer specific survival was defined as death for causes related to the ccRCC.
- Cohort 2; TCGA cohort: Progression free interval (PFI) was defined as 1 (= progressive patients) for patient having new tumor event, whether it was a progression of disease, local recurrence, distant metastasis, or died with the cancer without new tumor event, including cases with a new tumor event whose type is N/A. This PFI corresponds to what we call “disease progression”.
We agree that we cannot distinguish which patients had local relapse or distant metastasis for this TCGA cohort, but we do know who developed tumor progression, which is our endpoint.
Regarding the disease-specific survival (DSS) (or what we call cancer specific survival), the authors (Liu et al., 2018; Ref 12) defined as 1 for patient whose vital_status was Dead and tumor_status was WITH TUMOR. If a patient died from the disease shown in field of cause_of_death, the status of DSS would be 1 for the patient. 0 for patient whose vital_status was Alive or whose vital_status was Dead and tumor_status was TUMOR FREE. This is not a 100% accurate definition but is the best we could do with this dataset. Technically a patient could be with tumor but died of a car accident and therefore incorrectly considered as an event.
We have added TCGA definitions for Progression free interval (PFI) and disease-specific survival (DSS) in supplementary material Table S1.
Furthermore, since the authors (Liu et al., 2018; Ref 12) recommend taking DSS with caution, we have added this limitation to the study limitations (see lines 334-335).
Is the identified gene signature specific for ccRCC? Kindly also analyze TCGA-KIRP data to enhance the scientific merit of the study.
As mentioned for the reviewer, our signature is specific for ccRCC; both training and discovery datasets contains only clear cell renal carcinoma cases. We really appreciate the suggestion of the reviewer about TCGA-KIRP, but this is not the aim of the study and actually, we are conducting a parallel research project analyzing only papillary tumors.
For multivariate analyses related to individual markers, which clinical and pathological parameters were included? The multivariate data are shown only for the markers.
In our study univariate analysis was performed with individual markers; 637 overlapping genes (Table S1) and 3 clinical variables (pT stage, tumor size and ISUP grade). From this analysis, 132 genes and all clinical variables resulted significative (p<0.05). Then, LASSO regression was performed fixing the three clinical variables and using the 132 genes. LASSO analysis is a kind of multivariate linear regression analysis and allows us not only to select the prognosis factors but the genes that contributes more to the predictive value of the signature. As a result, we build a gene-based signature which comprise 6 genes (CERCAM, HS6ST2, ONECUT2, SOX12, TMEM132A and MIA2) and 3 clinicopathological variables (pT stage, tumor size and ISUP grade).
We have added this information in the revised version of the manuscript (see lines138 and 208-210)
It is unclear which cohorts were used for calculating Risk Score (RS) for disease progression. Only KM-plots were used for showing the clinical significance of the RS. It is unclear which data and which cohort was used to show disease progression-free survival and (B) cancer-specific survival in Figure 3. Multivariate analyses should be performed, as the increase in AUC is modest at best (0.82 versus 0.766). Please also include sensitivity and specificity values.
The risk score was computed taking into account the aforementioned 9 variables: 6 genes (CERCAM, HS6ST2, ONECUT2, SOX12, TMEM132A and MIA2) and 3 clinicopathological variables (pT stage, tumor size and ISUP grade). To evaluate the performance of this combined model, Kaplan Meier survival analysis and ROC was performed. We have included in the manuscript the sensitivity and the specificity of the threshold used to distinguish the 2 groups with different risk of progression (see lines 235-236).
TCGA cohort was used to show disease progression-free survival and cancer-specific survival in Figure 3. We have added this information in Figure 3 legend.

Round 2
Reviewer 3 Report
The authors describe their methods in a lot more details which makes the manuscript a lot more interpretable. I would recommend that the authors look into their univariate survival analysis, it seems like no correction was performed for multiple hypothesis testing and a p < 0.05 was used for selecting variables. There's is a likely hood of false positives creeping in here. Due to the lack of validation on other cohorts it's difficult to judge how generally applicable the survival model is, though if there is a lack of such datasets this is a reasonable compromise.
Author Response
Answers to comments of Reviewer #3
The authors describe their methods in a lot more details which makes the manuscript a lot more interpretable. I would recommend that the authors look into their univariate survival analysis, it seems like no correction was performed for multiple hypothesis testing and a p < 0.05 was used for selecting variables. There's is a likely hood of false positives creeping in here. Due to the lack of validation on other cohorts it's difficult to judge how generally applicable the survival model is, though if there is a lack of such datasets this is a reasonable compromise.
As stated by the reviewer, we did not use any correction in the univariate analysis. We agree with the reviewer that the correction for multiple hypothesis testing is the conventional approach to perform the univariate analysis. However, it must be considered that we did not perform the univariate analysis using the whole dataset. Instead, 637 selected genes which already showed to be differentially expressed between progressive and non-progressive patients in two different cohorts were used to perform the univariate analysis. Therefore, this univariate analysis represents a second variable selection to get the most prognostic-related genes.
Moreover, we did not want to be too restrictive in this univariate analysis to not narrow too much the genes selected for the next analysis, the LASSO regression. Considering, LASSO is a powerful statistical tool enable to reduce and select the genes to build the best predictive models, doing the corrections and limit more the number of genes would be inappropriate. Including more variables in LASSO regression ensures the selection of not correlated genes and more different combinations in order to select the most optimal model.
Regarding the lack of external validation, we agreed with the reviewer; however, as we explained previously an internal validation was performed.

Reviewer 4 Report
The authors provided response to my comments and wrote a lot. However, the response really did not address my concerns. For example, when asked to perform multivariate analysis they provide KM plots (this is not a statistical test - it just a representation - when the sample size is large p-value by log-rank or Wilcoxan test becomes significant). The authors skirt this topic in multiple responses.
Author Response
Answers to comments of Reviewer #4
The authors provided response to my comments and wrote a lot. However, the response really did not address my concerns. For example, when asked to perform multivariate analysis they provide KM plots (this is not a statistical test - it just a representation - when the sample size is large p-value by log-rank or Wilcoxan test becomes significant). The authors skirt this topic in multiple responses.
We are sorry that the reviewer feels that we have not address their concerns and we apologize for that.
Below, we show the results of the multivariate analysis of the model. If the reviewer considers we should add this analysis to the manuscript, please let us know.
Likelihood ratio test=95.36 on 9 df, p=<2.2e-16 n= 356, number of events= 68
|
|
hr |
|
95% |
|
C.I. |
|
P-val |
|
Size |
1.16 |
( |
1.07 |
- |
1.26 |
) |
0.0002 |
|
pT |
1.43 |
( |
1.00 |
- |
2.05 |
) |
0.0524 |
|
iSUP |
0.99 |
( |
0.66 |
- |
1.48 |
) |
0.9692 |
|
CERCAM |
0.99 |
( |
0.97 |
- |
1.01 |
) |
0.5482 |
|
CTAGE5 |
0.93 |
( |
0.89 |
- |
0.98 |
) |
0.0048 |
|
HS6ST2 |
1.42 |
( |
0.85 |
- |
2.37 |
) |
0.1792 |
|
ONECUT2 |
3.61 |
( |
1.95 |
- |
6.68 |
) |
0.0000 |
|
SOX12 |
1.04 |
( |
0.94 |
- |
1.16 |
) |
0.4219 |
|
TMEM132A |
1.06 |
( |
1.01 |
- |
1.12 |
) |
0.0191 |
Please, also let us know if there is any other question that is still not addressed, and we will do our best to response it.
